# Correlates of diabetes mellitus and hypertension in India: Change as evidenced from NFHS- 4 and 5 during 2015–2021

**Rishabh Kumar Rana**[1], **Ravi Ranjan Jha**[1], **Ratnesh Sinha**[2], **Dewesh Kumar**[3], **Richa Jaiswal**[4], **Urvish Patel**[5], **Jang Bahadur Prasad**[6], **Sitanshu Sekhar Kar**[7], **Sonu Goel**[8,9,10]*

1 Department of Community Medicine /PSM, Shaheed Nirmal Mahto Medical College and Hospital (SNMMCH), Dhanbad, Jharkhand, India, 2 Department of Community Medicine, Manipal Tata Medical College, Manipal Academy of Higher Education, Manipal, Karnataka, India, 3 Department of Community Medicine/PSM, Rajendra Institute of Medical Sciences (RIMS), Ranchi, Jharkhand, India, 4 Department of Emergency Medicine, Medical University of South Carolina, Charleston, South Carolina, United States of America, 5 Department of Public Health and Neurology, Icahn School of Medicine at Mount Sinai, New York, NY, United States of America, 6 Jawaharlal Institute of Postgraduate Medical Education and Research (JIPMER), Puducherry, India, 7 Department of Community Medicine, JIPMER, Puducherry, India, 8 Community Medicine and School of Public Health, PGI Chandigarh, Chandigarh, India, 9 Adjunct Associate Clinical Professor in the School of Medicine, Faculty of Education & Health Sciences, University of Limerick, Limerick, Ireland, 10 Faculty of Human and Health Sciences, Swansea University, Swansea, United Kingdom

* sonu.goel@ul.ie

**Data Availability Statement:** All .SAV files are available from the url https://dhsprogram.com/data/dataset/India_Standard-DHS_2020.cfm?flag=

## Abstract

Both diabetes mellitus (DM) and hypertension (HTN) have been on the rise in recent decades all over the world more remarkably in developing countries like India. We intend to measure the prevalence of DM and HTN in the Indian population and to compare the trends and various correlates of these diseases in NFHS (National Family Health Survey)-4 and 5. Data of NFHS-4 and 5 were accessed from dhs program website. All statistical analyses were done in SAS (version 9.4). Mixed effects survey logistic regression models were used for estimating odds ratio (OR).p-values <0.05 were considered significant .1,637,762 individual case entries were evaluated. Both the diseases showed an increasing trend as per the advancing age in both sexes. The highest prevalence of DM is seen in the age group of 45–49 years (7.8%) in females and > 50 years (11.9%) in males as per NFHS-5. Similarly, the highest prevalence of HTN was seen in the age group of45–49 years (31.2%) in females and > 50 years (41.4%) in males as per NFHS-5. The OR (95% CI) of prevalence of DM, HTN and both the diseases in age group >50 years was 14.46 (13.14–15.7), 16.65 (15.78–17.6), 79.5 (64.76–97.73) respectively when compared to reference age group15-19 years. Highest odds for having both DM, HTN concurrently was in age >50 years with aOR(95% CI) 65.32 (52.26–72.63) in NFHS 4 and 35.57 (97.47–45.53) in NFHS 5.Rise in prevalence of DM, HTN and concurrent presence is noted with an apparent increase in cases.

0 (accession numbers IAHR7ESV.ZIP, IAIR7ESV. ZIP, IAMR7ESV.ZIP . Whereas for NFHS 4 url is https://dhsprogram.com/data/dataset/India_ Standard-DHS_2015.cfm? flag=1 while accession numbers files are (IAHR74SV.ZIP, IAIR74SV.ZIP, IAMR74SV.ZIP).

**Competing interests:** No conflicting interest.

## Background

In recent decades, diabetes mellitus (DM) has risen to prominence as one of the most pervasive and devastating diseases of our day, with the potential to shorten one's life span, render them disabled, and drain their finances. The 9th edition of the International Diabetes Federation (IDF) reported in 2019 that the worldwide prevalence of DM has reached pandemic proportions, with 9% (463 million individuals) affected. Recent estimates for prevalence estimates for those aged 20–79 show that worldwide rates of DM will reach 10.5% (536.6 million) in 2021 and 12.2% (783.2 million) in 2045 [1, 2].

Hypertension (HTN) is a leading cause of disability-adjusted life years (DALYs) in men and women worldwide. Hemorrhagic and ischemic strokes and high systolic blood pressure are the biggest risk factors for ischemic heart disease [3]. HTN and DM are increasingly common and their interaction has been dubbed "bad companions." They mutually influence. Even with anti-HTN medication, uncontrolled blood pressure increases DM risk. Type 2 diabetics have a three-fold higher risk of HTN than non-diabetics. 50%–80% of type 2 DM patients have elevated blood pressure [4–6]. India is arguably the diabetic capital of the world, and cardiovascular diseases (CVDs) affected 14.1% of disability-adjusted life years (DALY) in 2016, up from 6.9% in 1990 [7, 8]. To establish an effective public health response and services system to manage the rise of these diseases, we need a clear understanding of the above scenario based on the local factors responsible for it. NFHS (National Family Health Survey) and DLHS (District level Household Survey) databases, which are indigenous and nationwide, have been studied in limited instances [8, 9].

The National Family Health Survey (NFHS) is an indigenous source of data for India, survey data for 2015–2016 NFHS 4 had 642344 female and 112122 male participants. According to NFHS 4 data, 11.3 percent of Indians have HTN, 13.6 percent of males and 8.8 percent of females aged 15–54 [7].

The urban-rural HTN prevalence gap is smaller than previously reported. As the lifestyle has changed over time so has the spread of HTN.

District Household Survey, a 2014 large sample database, is comparable to NFHS 4. Both surveys reported HTN prevalence at 25.3% (95% CI 25.0–25.6%), with men having a higher frequency (27.4%, CI 27.0–27.7%) than women (23.3–23.8%) [10]. The rule of halves is internationally known, yet most large data set studies, like the NFHS 4 study or the recently published 10,000-person National Noncommunicable Disease Monitoring Survey (NNDMS), have not shown its application in India. Amarchand R et al. (2022) used indigenous data from NFHS 4, DLHS 4, and AHS to find that 1.5% (0.76–2.96) of non-pregnant people had co- presence of high blood sugar and blood pressure [11].

India is progressing rapidly, thus we need more recent data for planning. Analyzing the data of NFHS and comparing the traits of HTN will enable us to better understand the changing pattern of the disease across the states, age groups, and other societal statuses. This information can be used to draw a state-wise prioritization map for a better-rationalized approach in the country. From recent larger country data, we intend to have a better picture of the coexistence of HTN and DM. Our objectives of the study are to know the sociodemographic traits of the survey population, know the prevalence of DM,HTN and their co-existence in this population. We also will look for correlates of the various factors which could be responsible for these conditions in the population surveyed.

## Methodology

In this study, secondary analysis was done of NFHS-4 and 5 data sets. These data setswere available in the public domain from the DHS Program website and were de-identified.[13]The

study was ethically approved by the Institute's Ethical Committee, Postgraduate Institute of Medical Education and Research (PGIMER), Chandigarh (IEC-08/2022-2535 dated 17.08.2022). We used the most recent round of fifth National Family Health Survey (NFHS-5), conducted in 2019–20, which collected data on India's population's health and nutrition. IIPS Mumbai coordinated this multi-round, nationwide survey for the Ministry of Health and Family Welfare. Information regarding several risk factors of HTN was added to the NFHS-5 survey, which was absent from the NFHS-4 survey [12].

The NFHS-5 survey used a stratified random sampling design, with two stages in both rural and urban regions. The first stage consisted of selecting villages (rural) and census enumeration blocks (urban), followed by selecting households at random in the second stage. The national report of the NHFS-5 provides thorough information on the specific sampling methodologies [12].

Data was collected from NFHS-5 individual recodes file (IAIR7AFL, which represented data for females) and the men's recodes file (IAMR7AFL). Total numbers of observations in individual and men recode files were 724115 and 101839 respectively. While for NFHS 4 the total numbers were for females as 699686 and for males 112122. In this study, we defined 'self-report diabetes' as all non-pregnant individuals who answered "yes" to the question "Do you currently have diabetes?". We defined 'undiagnosed diabetes' as those participants who answered 'no' to the question "do you currently have diabetes" and following a laboratory assessment either had an opportunistic fasting glucose level ≥126 mg/dL (referred to as 'fasting') or had a random glucose level ≥200 mg/dL (referred to as 'random'). Hypertension was defined as having a BP of 140/90 or responding "yes" to at least 1 of the 2 following questions: (a) "Were you told on 2 or more different occasions by a doctor or other health professional that you had hypertension or high blood pressure?" and (b) "To lower your blood pressure, are you now taking a prescribed medicine?"Concurrent presence of both in individuals who have both HTN and DM.

Independent variables studied were sex, age, religion, age groups (in 5-year bins), number of household members (total listed), occupation, wealth index (poorest, poor, middle, rich, and richest), level of education (none, primary, secondary, and higher), body mass index (bmi) (kg/m2), waist /hip ratio, smoking, drinking alcohol, place of residence (urban vs rural) and state of residence.While Comorbid conditions were self-reported heart disease, self-reported asthma and self-reported thyroid disorder.

All statistical analyses were performed by using the weighted survey methods in SAS (version 9.4) and p-value of <0.05 was considered to be significant. After assessing the parametricity of the data univariate analysis of differences between categorical variables was tested by using the chi-square test and analysis of differences between continuous variables (systolic and diastolic blood pressure, waist circumference, hip circumference, etc) was tested by using unpaired Student's t-test. Mixed-effects survey logistic regression models with weighted analysis were used for the categorical dependent variables, including HTN, DM, and concurrent presence of both HTN & DM for the outcome of interest, in order to estimate the odds ratio (OR) and 95% Confidence Interval for the association between various independent variables in the NFHS 4 and 5. Multivariate logistic regression was performed to estimate adjusted odds ratios with their 95% CI with every measurement, prevalence of HTN, DM, and concurrent presence of both HTN & DM as dependent variables and the above variables as independent.

## Results

It was noted that the burden of disease has increased in almost all the variables in NFHS-5 as compared to NFHS-4.A higher prevalence of DM, HTN, and Both diseases (HTN and DM) is

seen with the increase in age in both genders. This trend is shown in both NFHS-4 and NFHS-5 data also we can see that the prevalence of the disease is more in NFHS-5 data as compared to NFHS-4 data.The highest prevalence of DM is seen in the age group of 45–49 years in females and > 50 years in males, richest wealth index, BMI > 30, urban residents, widowed participants, and having a history of alcohol consumption (Table 1). A similar trend is shown for HTN and Both diseases (HTN and DM) (Tables 2 and 3). The overall prevalence of DM in females and males is 2% and 3.2% respectively as per NFHS-4 while in NHFS-5 it is 2.9% and 4.2% females and males respectively and an increase of 0.9% in females while 1% for males across the country (Table 1). Its prevalence rose from 13.2% to 13.5% in females and 18.7% to 21.6% in f males respectively as per NFHS 4 and NFHS 5 data (Table 2).

Co presence of DM and HTN in females grew from 0.8% to 1.0% while in males from 1% to 1.8%. The age group of 50–54 years in males was the major contributor to this morbid condition co-presence of both the non-communicable lifestyle diseases (Table 3). It was seen that mean ages for DM, HTN and co-presence of both was lower in NFHS-5 when compared with NFHS-4 with high statistical significance (S1, S2 Tables in S1 File).

Multivariate logistic regression analysis was performed for estimating the odds of having DM, HTN, and both depending on various factors as per Tables 1–3. Unadjusted Odds Ratio are presented in Table 4. Odds of having DM, HTN or their co-presence were seen to be less if one resided in rural areas and as one got educated. Higher odds of having these diseases were found to be in these predictors a) Age group as age grew when compared with 15–19 age group the risk of having these conditions grew with max in the high age group of 50–54. These odds were following the same pattern in both NFHS-4 and NFHS-5 and very high statistical significance. As body mass index grows and crosses 25 odds start to increase for the occurrence of these diseases and their co-presence. Waist Hip Ratio an addition for NFHS-5 follows the accepted norm of having higher odds of having DM, HTN or both. When compared to Waist Hip Ratio < = 1 the odds are 2.76 times more of having the co-presence of DM and HTN. Seen in both NFHS-4 and NFHS-5 is a protective trend in the case of education as compared to uneducated. As one tends to get a higher degree the odds are on the lower side for the occurrence of these conditions. Marriage is an important predictor for the presence of these conditions when compared to never married. It was seen that the highest odds of having these conditions arise when one is either married or widowed. This trend was seen both in NFHS-4 and NFHS-5. Usage of cigarettes, khaini, and alcohol when compared to non-smokers and non-drinkers were strong predictors for these conditions. The chances of having any of these three conditions increased drastically when patients had self-reported heart disease, asthma, thyroid, or cancer.

Logistic regression analysis was applied to adjust age with various variables for calculating the adjusted Odds Ratio for the occurrence of DM, HTN and their co-presence (Table 5). As noted, education seems to have an odds ratio showing more education may be less related with these diseases, e.g. Both odds were 0.43(95%CI 0.39–0.47), however when adjusted for age, education followed a similar trend of greater risks getting both HTN and DM aOR 1.40 (95% CI 1.29–1.53).

## State-wise prevalence heat map in India according to overall prevalence

A state-wise heat map representing existing prevalence and earlier recorded prevalence from NFHS-5 and NFHS-4 across the states of all the diseases separately and their co-presence is shown in S1-S3 Figs in S1 File. The difference in the prevalence of HTN across the states was statistically significant as per unpaired T test with p < .05 while for DM and presence of both HTN and Type 2 DM, this difference was not of statistical significance. As per NFHS-5 data,

**Table 1. Study participant details having only DM as per NFHS 4 and NFHS 5 survey.**

| | NFHS 4 | | NFHS 5 | |
|---|---|---|---|---|
| | Diabetes N = 16934(2.2) | | Diabetes N = 23896(3.1) | |
| | Female | Male | Female | Male |
| **Age** | | | | |
| 15–19 | 429(0.4) | 103(0.6) | 1031(0.9) | 176(1.1) |
| 20–24 | 644(0.5) | 120(0.8) | 1260(1.1) | 179(1.4) |
| 25–29 | 983(0.9) | 201(1.3) | 1778(1.6) | 233(1.8) |
| 30–34 | 1478(1.6) | 311(2.2) | 2219(2.3) | 346(2.8) |
| 35–39 | 2261(2.6) | 460(3.5) | 3371(3.7) | 556(4.7) |
| 40–44 | 3256(4.4) | 606(5.3) | 4240(5.6) | 621(6.3) |
| 45–49 | 4506(6.5) | 767(7.2) | 6077(7.8) | 883(8.9) |
| = >50 | 0 | 809(9.9) | 0 | 926(11.9) |
| Total | 13557(2.0) | 3377(3.2) | 19976(2.9) | 3920(4.20) |
| **Wealth Index** | | | | |
| Poorest | 1327(1.0) | 308(1.7) | 2817(2.2) | 467(2.7) |
| Poorer | 1859(1.3) | 466(2.1) | 3718(2.6) | 694(3.6) |
| Middle | 2533(1.8) | 645(2.8) | 4077(2.9) | 800(4.2) |
| Richer | 3640(2.7) | 874(3.9) | 4396(3.7) | 862(4.6) |
| Richest | 4198(3.3) | 1084(5.0) | 4968(3.7) | 1097(5.9) |
| **Level of Education** | | | | |
| None | 4123(2.2) | 389(2.7) | 5391(3.40) | 452(4.0) |
| Incomplete Education | 0 | 0 | 2897(3.60) | 488(4.50) |
| Incomplete Primary | 980(2.5) | 230(3.3) | – | - |
| Primary | 1075(2.3) | 207(3.0) | – | - |
| Incomplete Secondary | 5084(1.9) | 1474(2.9) | 9144(2.80) | 2183(4.30) |
| Complete Secondary | 962(1.6) | 402(3.3) | 355(1.50) | 96(2.50) |
| Higher | 1333(1.7) | 675(4.1) | 2189(2.30) | 701(4.3) |
| **BMI** | | | | |
| <18.5 | 1057(0.7) | 277(1.3) | 1500(1.3) | 275(2.0) |
| 18.5–22.9 | 3462(1.1) | 1043(2.1) | 5874(1.9) | 1113(2.7) |
| 23–24.9 | 1938(2.3) | 652(2.9) | 3092(3.2) | 779(4.8) |
| 25.0–27.4 | 2434(4.0) | 640(5.7) | 3310(4.6) | 786(6.6) |
| 27.5–29.9 | 1802(5.7) | 392(8.3) | 2303(6.1) | 433(8.5) |
| >30. | 2428(8.6) | 349(12.2) | 3261(9.1) | 399(12.0) |
| **Residence** | | | | |
| Urban | 5914(3.0) | 1371(4.1) | 6685(4.0) | 1341(5.7) |
| Rural | 7643(1.6) | 2006(2.7) | 13291(2.6) | 2579(3.7) |
| **Marital Status (Currently in Union)** | | | | |
| Never | 846(0.5) | 377(1.0) | 1777(1.1) | 512(1.5) |
| Married | 11687(2.4) | 2941(4.4) | 16737(3.5) | 3330(5.7) |
| Widowed | 810(4.2) | 37(4.6) | 1157(5.6) | 44(6.5) |
| Divorced | 71(2.4) | 8(2.4) | 102(3.9) | 15(4.4) |
| No Longer Living Together with Partner | 143(3.2) | 14(3.4) | 203(4.0) | 19(4.6) |
| **Tobacco use** | | | | |
| Cigarette | 72(3.3) | 618(3.8) | 58(4.5) | 666(5.0) |
| **Number of Cigarettes in last 24 hours** | | | | |
| 1–4 | 45(3.0) | 257(3.1) | 49(4.7) | 408(4.7) |
| 5–10 | 15(3.9) | 123(4.3) | 5(3.45) | 174(6.1) |

*(Continued)*

**Table 1.** (Continued)

| | NFHS 4 | | NFHS 5 | |
|---|---|---|---|---|
| | **Diabetes N = 16934(2.2)** | | **Diabetes N = 23896(3.1)** | |
| | **Female** | **Male** | **Female** | **Male** |
| >10 | 12(4.5) | 135(4.7) | 4(4.2) | 84(4.9) |
| Chewing Tobacco | 264(1.9) | 436(2.9) | 146(3.0) | 94(4.55) |
| No smoke | 13485(2.0) | 2759(3.0) | 19918(2.9) | 3254(4.1) |
| **Alcohol Use** | | | | |
| Drinks alcohol | 322(2.0) | 1262(3.8) | 372(2.9) | 1217(5.0) |
| Not drinks alcohol | 13235(2.0) | 2115(2.9) | 19604(2.90 | 2703(3.9) |
| **Waist Hip Ratio NFHS 5** | | | | |
| ≤1 | - | - | 18852(2.8) | 3217(3.6) |
| >1 | - | - | 177(4.2) | 84(10.6) |

- Percentages in parentheses
- - Data Not available

Sikkim state had the highest total prevalence of HTN with 32.21% which is an increase of 7.90% from NFHS-4, while the states of Puducherry and Kerala were having the highest prevalence percentage of DM at 6.20% and 6.60% respectively in NFHS-5. Kerala had an increase of 0.5% whereas Puducherry had an increase of 0.70% with respect to NFHS-4. As per NFHS-5 highest prevalence for presence of both HTN and DM was observed in Chandigarh >3.0%. Chandigarh noted an increase of 2.10% from NFHS-4.

Various co-morbid conditions were reported by the participants, we plotted a bar graph of these conditions. In NFHS-5 chronic kidney disease was introduced among the questions Self-Reported thyroid disease and self-reported heart diseases were the major comorbid conditions reported by respondents while the absolute numbers of such conditions were markedly more in hypertensives. Percentage-wise self-reported asthma increased both in hypertensive females and males are shown in S4, S5 Figs in S1 File. The number of such diseases reported rose in NFHS-5 as compared to NFHS-4. S3 Table in S1 File shows the multivariate logistic regression applied on self-reported disorders in NFHS-4 and 5. In self-reported cancer, the odds were seen to be as high as 37 times to 20 times for predicting the presence of DM, the trend of cancer being a strong predictor was consistent in NFHS-4 and NFHS-5.

## Discussion

To our knowledge in our study is the first study to compare the findings of NFHS-4 and NFHS-5 for the whole country data. We have attempted to analyze and compare the findings of NFHS-4 and NFHS-5 data bases looking for overall prevalence of HTN, DM and concurrent presence of both HTN and DM together. More than 16 lakhs data samples were analyzed for finding the various correlates responsible for odds of these diseases across the country. The present study has mapped state wise prevalence of the above mentioned morbidities. Results of the study shows the trend of the disease in relationship with predictors of disease such as gender, age, residence, literacy, wealth index, BMI, waist hip ratio etc to ascertain their relationship and role.

### 1. Summary of key findings of the study

We found that the prevalence of HTN, DM, and the concurrent presence of both has increased in India. This is the latest comparison of both country-wide exhaustive data sets in the Indian

**Table 2. Study participant details having only HTN as per NFHS 4 and NFHS 5 survey.**

| | NFHS 4 | | NFHS 5 | |
|---|---|---|---|---|
| | Hypertension N = 101790(14.0) | | Hypertension N = 110245(14.5) | |
| | Female | Male | Female | Male |
| **Age** | | | | |
| 15–19 | 4469(3.9) | 855(4.9) | 4517(3.9) | 741(5.2) |
| 20–24 | 6522(5.99) | 1425(9.5) | 6162(5.6) | 1317(10.7) |
| 25–29 | 8949(8.7) | 2055(14.2) | 9098(8.3) | 1828(15.1) |
| 30–34 | 11490(13.3) | 2545(19.4) | 11729(12.5) | 2443(21.8) |
| 35–39 | 15007(18.6) | 2976(23.9) | 16502(18.1) | 2911(26.2) |
| 40–44 | 16714(24.4) | 3045(28.3) | 18889(24.9) | 3028(32.3) |
| 45–49 | 19736(30.3) | 3254(32.5) | 24543(31.2) | 3438(36.2) |
| = >50 | 0 | 2748(35.7) | 0 | 3120(41.4) |
| Total | 82887(13.2) | 18903(18.7) | 91440(13.5) | 18826(21.6) |
| **Wealth Index** | | | | |
| Poorest | 13695(11.1) | 2338(13.8) | 16539(12.7) | 3172(19.2) |
| Poorer | 16583(12.2) | 3258(15.4) | 18655(13.1) | 3580(19.7) |
| Middle | 17090(13.0) | 4014(18.3) | 18584(13.4) | 3821(21.3) |
| Richer | 17901(14.6) | 4466(21.6) | 18395(13.7) | 3959(22.5) |
| Richest | 17618(15.3) | 4827(23.9) | 19267(14.8) | 4294(25.1) |
| **Level of Education** | | | | |
| None | 29831(16.6) | 2706(19.9) | 29856(19.1) | 2598(24.40) |
| Incomplete Education | 0 | 0 | 13632(17.10) | 2530(24.70) |
| Incomplete Primary | 6140(16.6) | 1410(21.5) | - | - |
| Primary | 6200(14.4) | 1213(18.7) | - | - |
| Incomplete Secondary | 28219(11.5) | 8060(17.0) | 37650(11.70) | 9971(20.80) |
| Complete Secondary | 5450(10.1) | 2149(18.7) | 1574(6.70) | 478(13.20) |
| Higher | 7047(10.1) | 3365(21.7) | 8707(9.40) | 3249(21.70) |
| **BMI** | | | | |
| <18.5 | 10037(7.1) | 1685(8.5) | 8107(7.0) | 1220(9.4) |
| 18.5–22.9 | 27954(9.8) | 6922(14.4) | 29410(9.7) | 6211(16.2) |
| 23–24.9 | 12534(16.4) | 3794(24.5) | 14813(15.7) | 3768(25.1) |
| 25.0–27.4 | 12616(22.8) | 3416(32.8) | 14774(20.8) | 3662(33.1) |
| 27.5–29.9 | 8148(28.2) | 1833(41.6) | 9852(26.4) | 1971(41.2) |
| >30. | 9124(35.8) | 1180(44.2) | 11550(32.7) | 1421(44.6) |
| **Residence** | | | | |
| Urban | 25900(14.4) | 6726(21.4) | 24342(14.8) | 5450(24.6) |
| Rural | 56987(12.7) | 12177(17.5) | 67098(13.1) | 13376(20.5) |
| **Marital Status (Currently in Union)** | | | | |
| Never | 8294(5.3) | 3354(9.2) | 8777(5.2) | 3224(10.4) |
| Married | 69149(15.5) | 15172(24.0) | 75951(15.9) | 15189(27.7) |
| Widowed | 4330(23.8) | 209(27.5) | 5400(26.0) | 223(33.7) |
| Divorced | 490(17.7) | 82(25.6) | 444(17.0) | 77(24.1) |
| No Longer Living Together with Partner | 624(14.7) | 86(21.7) | 868(16.8) | 113(27.6) |
| **Tobacco use** | | | | |
| Cigarette | 343(16.8) | 3204(20.8) | 242(18.5) | 3007(24.4) |
| **Number of Cigarettes in last 24 hours** | | | | |
| 1–4 | 220(16.2) | 1529(19.3) | 192(18.2) | 1922(23.6) |
| 5–10 | 66(17.6) | 572(21.8) | 32(20.6) | 672(26.3) |
| >10 | 52(20.6) | 657(24.3) | 18(18.8) | 413(25.60) |

*(Continued)*

**Table 2.** (Continued)

| | NFHS 4 | | NFHS 5 | |
|---|---|---|---|---|
| | Hypertension N = 101790(14.0) | | Hypertension N = 110245(14.5) | |
| | Female | Male | Female | Male |
| Chewing Tobacco | 2453(18.2) | 2943(20.80 | 823(16.6) | 510(25.6) |
| No smoke | 82544(13.2) | 15699(18.3) | 91198(13.5) | 15189(21.1) |
| **Alcohol Use** | | | | |
| Drinks alcohol | 3226(22.1) | 7509(23.9) | 2988(23.1) | 6760(29.2) |
| Not drinks alcohol | 79661(13.0) | 11394(16.4) | 88452(13.4) | 12066(18.8) |
| **Waist Hip Ratio NFHS 5** | | | | |
| < = 1 | - | - | 88013(13.4) | 16647(20.2) |
| >1 | - | - | 660(15.9) | 271(36.0) |

context. The increase in prevalence is more marked in the age group of 30–50. This suggests that all cases who are coming for DM or HTN should be screened for both, either in a diabetic clinic or a clinic for heart care.

## 2. Comparison of diabetes in the database of NFHS-4 and NFHS-5

The prevalence of DM was 2.2 and 3.1 as per NFHS-4 and NFHS-5 data respectively showing an increasing trend. In NFHS-5 the prevalence and risk of DM were seen more in males, age group > 50 years, richest wealth index, people having incomplete education, BMI > 30 kg/m2, urban residents, widowed, smokers, alcoholics and WHR > 1.The highest prevalence of DM across the states was seen in Puducherry (6.6). We noted an increasing trend regarding odds of having DM as the individual aged. The adjusted odds for age with other factors also had similar trend. Other studies about NFHS-5 also had similarity regarding the prevalence of DM in old aged individuals [13]. Such trends of overall prevalence in these states were seen earlier in NFHS-4 data analysis by Geldsetzer, Pascal et al earlier [8]. As noted in our study we noted higher odds of having DM for people living in urban areas these odds were similar even after adjustment. Large studies done in Indian context have reported similar findings [14]. Marital status and association of DM was seen to be statistically associated in cases of separated individuals, the trends in NFHS-4 and NFHS-5 were statistically significant when odds were calculated. The factors have been highlighted by previous study done in India. Possible explanation has generally been given as lack of care and stress of living alone, and sleep deprivation. Other studies have established possible mechanism for this strong association [15]. Odds of having DM was also related with educational status and it was higher in higher study groups. Earlier studies have reported association of DM in Iran with education in Asian context [16]. Possible explanation has been nature of employment which is generally associated with higher education, higher income and sedentary life style. Similarly higher wealth index, use of tobacco and consumption of alcohol were seen to be strong risk factors for occurrence of DM. Studies in the past have highlighted the mechanism by which alcohol tends to increase the sugar levels and thus by worsening the DM [17].

## 3. Comparison of HTN in the database of NFHS-4 and NFHS-5

HTN prevalence was 14 and 14.5 as per NFHS-4 and NFHS-5 data respectively. NFHS-5 data reveals that the prevalence and risk of HTN has increased in males, age group >50 years, richest wealth index, people having no education, BMI > 30 kg/m2, urban residents, widowed,

**Table 3. Study participant details having both HTN and DM as per NFHS 4 and NFHS 5.**

| | NFHS 4 | | NFHS 5 | |
|---|---|---|---|---|
| | Both HTN & DM N = 5936(0.8) | | Both HTN & DM N = 8235(1.1) | |
| | Female | Male | Female | Male |
| **Age** | | | | |
| 15–19 | 26(0.0) | 5(0.9) | 99(1.5) | 15(0.1) |
| 20–24 | 61(0.1) | 15(0.1) | 140(2.1) | 35(0.3) |
| 25–29 | 162(0.2) | 41(0.3) | 286(4.3) | 34(0.3) |
| 30–34 | 360(0.4) | 85(0.7) | 514(7.7) | 101(0.9) |
| 35–39 | 750(1.0) | 158(1.3) | 1128(16.9) | 192(1.8) |
| 40–44 | 1300(1.9) | 246(2.3) | 1691(25.3) | 269(3.0) |
| 45–49 | 1978(3.1) | 357(3.6) | 2824(42.3) | 429(4.7) |
| = >50 | 0 | 392(5.2) | 0 | 478(6.7) |
| Total | 4637(0.8) | 1299(1.3) | 6682(1.0) | 1553(1.8) |
| **Wealth Index** | | | | |
| Poorest | 316(0.3) | 84(0.5) | 778(0.6) | 141(0.9) |
| Poorer | 547(0.4) | 138(0.7) | 1169(0.9) | 253(1.4) |
| Middle | 822(0.6) | 209(1.0) | 1376(1.0) | 302(1.7) |
| Richer | 1356(1.1) | 394(1.9) | 1506(1.2) | 357(2.1) |
| Richest | 1596(1.4) | 474(2.4) | 1853(1.5) | 500(3.0) |
| **Level of Education** | | | | |
| None | 1515(0.9) | 138(1.0) | 1963(1.30) | 173(1.70) |
| Incomplete Education | 0 | 0 | 1132(1.50) | 211(2.10) |
| Incomplete Primary | 357(1.0) | 98(1.5) | - | - |
| Primary | 406(1.0) | 86(1.3) | - | - |
| Incomplete Secondary | 1677(0.7) | 559(1.2) | 2940(0.90) | 852(1.80) |
| Complete Secondary | 298(0.6) | 137(1.2) | 82(.40) | 28(0.80) |
| Higher | 384(0.6) | 281(1.8) | 565(0.60) | 289(2.00) |
| **BMI** | | | | |
| <18.5 | 162(0.1) | 41(0.2) | 229(0.2) | 53(0.4) |
| 18.5–22.9 | 773(0.3) | 302(0.6) | 1274(0.4) | 333(0.9) |
| 23–24.9 | 657(09) | 228(1.5) | 975(1.1) | 313(2.1) |
| 25.0–27.4 | 912(1.7) | 311(3.0) | 1263(1.8) | 362(3.4) |
| 27.5–29.9 | 794(2.8) | 226(5.2) | 1041(2.9) | 218(4.8) |
| >30. | 1205(4.8) | 184(7.0) | 1670(4.9) | 218(7.25) |
| **Residence** | | | | |
| Urban | 6726(21.4) | 2225(1.3) | 2515(1.6) | 607(2.8) |
| Rural | 12177(17.5) | 2412(0.5) | 4167(0.8) | 946(1.5) |
| **Marital Status (Currently in Union)** | | | | |
| Never | 137(0.1) | 61(0.2) | 254(0.2) | 87(0.3) |
| Married | 4092(0.9) | 1214(1.9) | 5811(1.3) | 1430(2.7) |
| Widowed | 350(2.0) | 16(2.1) | 510(2.6) | 25(4.0) |
| Divorced | 24(0.9) | 3('1.0) | 39(1.5) | 6(1.9) |
| No Longer Living Together with Partner | 34(0.8) | 5(1.3) | 68(1.4) | 5(1.3) |
| **Tobacco use** | | | | |
| Cigarette | 24(1.3) | 233(1.5) | 22(1.8) | 256(2.2) |
| **Number of Cigarettes in last 24 hours** | | | | |
| 1–4 | 13(1.0) | 103(1.3) | 19(1.9) | 160(2.0) |
| 5–10 | 6(1.7) | 40(1.6) | 1(0.7) | 63(2.6) |
| >10 | 5(2.1) | 49(1.9) | 2(2.2) | 33(2.2) |

(*Continued*)

**Table 3.** (Continued)

| | NFHS 4 | | NFHS 5 | |
|---|---|---|---|---|
| | **Both HTN & DM N = 5936(0.8)** | | **Both HTN & DM N = 8235(1.1)** | |
| | **Female** | **Male** | **Female** | **Male** |
| Chewing Tobacco | 103(0.80 | 171(1.2) | 57(1.2) | 42(2.20) |
| No smoke | 4613(0.8) | 1066(1.3) | 6660(1.0) | 1297(1.8) |
| **Alcohol Use** | | | | |
| Drinks alcohol | 102(0.7) | 530(1.7) | 131(1.1) | 546(2.5) |
| Not drinks alcohol | 4535(0.8) | 769(1.1) | 6551(1.0) | 1007(1.6) |
| **Waist Hip Ratio NFHS-5** | | | | |
| < = 1 | - | - | 6153(1.0) | 1167(1.5) |
| >1 | - | - | 70(2.0) | 51(7.1) |

smokers, alcoholics and WHR > 1. Prevalence for HTN has increased by 7 in Sikkim while in Nagaland it has decreased when compared with NFHS-4 data.

Other authors like Vennu et al. have raised this issue in their NFHS-4 and other analysis involving large country data set [18, 19]. In our study overall prevalence of HTN in NFHS-5 was 14.5 while for male it was 21.6 and females it was 13.5. This prevalence was more markedly increased in male with a rise of approx. 3. Of note in the various states about the prevalence of HTN in Kerala as it has been noted that despite the epidemiological transition in NFHS-4 the state had a lower prevalence as compared to other states [20]. The percentage of prevalence has risen from 11.5 to 13.5 in NFHS-5 possible explanations could be the large number of female participants and age less than 60 years. As noted by Ghosh et al hypertension in India is no longer a disease afflicting mostly the rich, but now the poor are also having a higher prevalence of HTN [19]. Percentages of prevalence in the poor have increased in NFHS-4 as compared to NFHS-5. We noted educational attainment to be inversely proportional to the prevalence of HTN. This was also noted by Gosh et al in their NFHS-4 analysis and in NFHS-5 trend is same. Persons residing in rural areas had lesser chances of getting HTN as the odds were lesser with statistical significance. This was also seen in NFHS-4 data by Gosh et al. [19]. Obesity and alcohol have strong relationship with HTN as seen in both NFHS-4 and NFHS-5 and this similar relationship of these variables in Indian large population studies have been reported earlier by Bhansali et al. [20]. We noted strong association of smokers and tobacco users with HTN prevalence. Ghosh et al had little different finding then us in their NFHS-4 data analysis. However, our findings are consistent with available evidence from various Indian studies which has attributed cigarette smoking, tobacco use and obesity as the top contributors apart from age for HTN [21].

## 4. Comparison of concurrent presence of both DM and HTN together in the database of NFHS-4 and NFHS-5

Bischops, Anne C et al in their large study found overall prevalence of HTN to be 22.1 while that for raised blood sugar was 6.4 and prevalence of having concurrently raised blood sugar and HTN was 1.5 [10]. In NFHS 5 we found the concurrent occurrence of DM and HTN to be 1.1. Tripathy et al. in their study found an estimated prevalence of 4.5 which was way higher than our findings [22]. We can say that our findings are close enough when considered with the findings of Bischops et al. [10]. We noted higher concurrent prevalence of DM and HTN in older age, high BMI group individuals, more in individuals having higher waist hip ration >1, having higher wealth index and having higher education. Corsi et al in their analysis of

**Table 4. Multivariate regression analysis for obtaining odds of various correlates with the occurrence of either DM, HTN or both concurrently as per NFHS-4 and NFHS-5.**

| | Diabetes Values as OR (95CI) *denotes p value < .05 | | Hypertension Values as OR (95CI) *denotes p value < .05 | | Both (Values as OR (95CI) *denotes p value < .05 | |
|---|---|---|---|---|---|---|
| | NFHS 4 | NFHS 5 | NFHS 4 | NFHS 5 | NFHS 4 | NFHS 5 |
| **Age Groups** | | | | | | |
| 15–19 | 1 | 1 | 1 | 1 | 1 | 1 |
| 20–24 | 1.47* (1.32–1.65) | 1.23* (1.14–1.33* | 1.60* (1.55–1.66) | 1.53* (1.47–1.58) | 2.57* (1.69–3.91) | 1.61* (1.27–2.04) |
| 25–29 | 2.45* (2.21–2.71) | 1.73* (1.61–1.86) | 2.48* (2.4–2.56) | 2.32* (2.24–2.40) | 7.44* (5.10–10.87) | 2.98* (2.41–3.69) |
| 30–34 | 4.38* (3.97–4.82) | 2.59* (2.42–2.78) | 3.92* (3.79–4.05) | 3.69* (3.56–3.80) | 19.26* (13.38–27.73) | 6.71* (5.50–8.20) |
| 35–39 | 7.22* (6.58–7.93) | 4.17* (3.91–4.45) | 5.72* (5.54–5.91) | 5.52* (5.34–5.7) | 42.36* (29.61–60.6) | 14.98* (12.37–18.15) |
| 40–44 | 12.33* (11.25–13.5) | 6.38* (5.98–6.80) | 7.92* (7.76–8.17) | 8.17* (7.92–8.43) | 85.53* (59.93–122.0) | 27.20* (22.52–32.86) |
| 45–49 | 18.23* (16.67–19.9) | 9.11* (8.57–9.69) | 10.52* (10.2–10.85) | 10.95* (10.62–11.3) | 138.40* (97.09–197.2) | 44.43* (36.85–53.57) |
| >50 | 28.37* (25.37–31.7) | 14.36* (13.14–15.7) | 13.26* (12.56–14.0) | 16.65* (15.78–17.6) | 230.61* (159.6–332.7) | 79.55* (64.76–97.73) |
| **BMI** | | | | | | |
| <18 | 1 | 1 | 1 | 1 | 1 | 1 |
| 18.5–22.9 | 1.62* (1.52–1.72) | 1.49* (1.42–1.57) | 1.49* (1.46–1.52) | 1.49* (1.46–1.53) | 2.58* (2.22–2.99) | 2.15* (1.90–2.43) |
| 23–24.9 | 3.36* (3.14–3.59) | 2.60* (2.46–2.75) | 2.75* (2.68–2.82) | 2.61* (2.54–2.68) | 7.75* (6.65–9.03) | 5.42* (4.78–6.15) |
| 25.0–27.4 | 5.62* (5.26–5.99) | 3.73* (3.53–3.95) | 4.09* (3.99–4.20) | 3.71* (3.62–3.81) | 15.08* (13.01–17.5) | 9.27* (8.19–10.5) |
| 27.9–29.9 | 8.07* (7.53–8.65) | 4.93* (4.65–5.24) | 5.46* (5.29–5.62) | 4.99* (4.84–5.14) | 25.23* (21.69–29.35) | 14.15* (12.46–16.07) |
| 30 or above. | 12.39* (11.6–13.24) | 7.46* (7.04–7.90) | 7.36* (7.13–7.58) | 6.49* (6.30–6.69) | 41.43* (35.74–48.03) | 23.76* (21.01–26.87) |
| **Waist /Hip Ratio for NFHS-5** | | | | | | |
| <1 | - | 1 | - | 1 | - | 1 |
| >1 | - | 1.82* (1.60–2.06) | - | 1.42* (1.33–1.53) | - | 2.78* (2.33–3.31) |
| **Residence** | | | | | | |
| Rural | 0.54* (0.52–0.55) | 0.56* (0.47–0.66) | 0.84* (0.83–0.85) | 0.85* (0.84–0.86) | 0.44* (0.42–0.47) | 0.52* (0.50–0.54) |
| Urban | 1 | 1 | 1 | 1 | 1 | 1 |
| **Education** | | | | | | |
| None | 1 | 1 | 1 | 1 | 1 | 1 |
| Incomplete Education | - | 1.07* (1.03–1.12) | - | 0.98 (0.85–1.14) | - | 1.06 (0.98–1.14) |
| Primary incomplete | 1.19* (1.12–1.27) | - | 1.03 (1.01–1.06) | - | 1.23* (1.11–1.37) | - |
| Primary complete | 1.10* (1.04–1.17) | - | 0.87* (0.84–0.89) | - | 1.16 (1.05–1.28) | - |
| Secondary incomplete | 0.94* (0.91–0.98) | 0.86* (0.83–0.89) | 0.69* (0.68–0.70) | 0.86* (0.76–0.97) | 0.89* (0.83–0.95) | 0.67* (0.64–0.71) |
| Secondary Complete | 0.86* (0.81–0.92) | 0.46* (0.42–0.51) | 0.64* (0.63–0.66) | 0.59* (0.44–0.79) | 0.77* (0.69–0.85* | 0.25* (0.20–0.31) |
| Higher | 0.98 (0.92–1.03) | 0.74* (0.71–0.78) | 0.68* (0.66–0.69) | 0.81* (0.69–0.94) | 0.94* (0.82–0.99) | 0.43* (0.39–0.47) |

*(Continued)*

**Table 4.** (Continued)

| | Diabetes Values as OR (95CI) *denotes p value < .05 | | Hypertension Values as OR (95CI) *denotes p value < .05 | | Both (Values as OR (95CI) *denotes p value < .05 | |
|---|---|---|---|---|---|---|
| | NFHS 4 | NFHS 5 | NFHS 4 | NFHS 5 | NFHS 4 | NFHS 5 |
| **Marital Status** | | | | | | |
| Never | 1 | 1 | 1 | 1 | 1 | 1 |
| Married | 4.5* (4.24–4.77) | 3.36* (3.22–3.51) | 3.07* (3.01–3.13) | 2.39* (2.16–2.66) | 10.22* (8.86–11.78) | 5.67* (3.54–9.08) |
| Widowed | 7.17* (6.56–7.84) | 5.22* (4.86–5.60) | 4.89* (4.71–5.08) | 3.31* (2.42–4.54) | 19.35* (16.27–23.02) | 11.04* (4.89–24.93) |
| Divorced | 4.09* (3.24–5.14) | 3.62* (2.99–4.37) | 3.53* (3.21–3.87) | 1.92* (1.23–3.01) | 8.77* (5.86–13.14) | 1.58 (0.20–12.00) |
| No longer living together | 5.40* (4.56–6.39) | 3.65* (3.17–4.20) | 2.81* (2.59–3.05) | 2.27* (1.41–3.66) | 8.32* (5.89–11.47) | 5.12* (1.16–22.50) |
| **Wealth Quintile** | | | | | | |
| Poorest | 1 | 1 | 1 | 1 | 1 | 1 |
| Poorer | 1.25* (1.17–1.33) | 1.23* (1.17–1.28) | 1.19* (1.09–1.14) | 1.06 (0.94–1.20) | 1.53* (1.35–1.73) | 1.64* (1.07–2.51) |
| Middle | 1.73* (1.63–1.84) | 1.39* (1.22–1.45) | 1.23* (1.20–1.26) | 1.25* (1.10–1.42) | 2.36* (2.10–2.65) | 2.04* (1.34–3.12) |
| Richer | 2.64* (2.50–2.80) | 1.55* (1.48–1.62) | 1.43* (1.40–1.46) | 1.33* (1.17–1.51) | 4.31* (3.85–4.80) | 1.85* (1.18–2.88) |
| Richest | 3.29* (3.11–3.48) | 1.83* (1.75–1.91) | 1.53* (1.50–1.57) | 1.48* (1.29–1.69) | 5.38* (4.83–5.99) | 3.51* (2.32–5.30) |
| **Tobacco Use** | | | | | | |
| Cigarette | 1.76* (1.63–1.90) | 1.65* (1.53–1.78) | 1.59* (1.53–1.65) | 1.16* (1.04–1.17) | 1.88* (1.65–2.13) | 1.04 (.88–1.23) |
| Khaini | 1.13* (1.05–1.22) | 1.27* (0.99–1.28) | 1.52* (1.47–1.57) | 1.17* (1.10–1.25) | 1.26* (1.11–1.42) | 1.18 (0.97–1.45) |
| No smoke | 1 | 1 | 1 | 1 | 1 | 1 |
| **Alcohol Use** | | | | | | |
| Drinks alcohol | 1.53* (1.45–1.61) | 1.44* (1.36–1.51) | 1.97* (1.93–2.02) | 2.29* (2.23–2.34) | 1.79* (1.65–1.95) | 1.83* (1.69–1.98) |
| Not drinks alcohol | 1 | 1 | 1 | 1 | 1 | 1 |

NFHS 4 had similar results [23]. Large data set studies of national representative population in similar settings have revealed the same predictors regarding the concurrent presence of HTN and DM [24]. The rise in prevalence of DM is also reflected with rise in prevalence of Gestational diabetes as noted by Basu S et al in their study on NFHS 5 [25].

Our findings about the prevalence of DM, HTN and presence of both had interesting findings state wise, as we noted relatively higher prevalence of HTN in the north easter states of Sikkim (Highest in India), Arunachal Pradesh and Nagaland in both NFHS 4 and 5.

Older age groups were key vulnerable groups which were seen to have more prevalence of either DM, HTN, or concurrent DM and HTN.

## 5. Pathological basis for correlation between co presence of HTN along with DM and its ramifications

Recent studies have shown that hypertension can lead to diabetes in the same individual. Song et al. [26] in their cohort study of 3017 Japanese participants were unique in forecasting that long-term blood pressure/HTN is a risk factor for type 2 diabetes. Tanaka et al. [27] have

**Table 5. Multivariate logistic regression analysis for various variables with adjusted OR like age, BMI and marital status with 95 CI for having HTN, diabetes and both of NFHS-4 and NFHS -5.**

| | NFHS 4 | NFHS 5 | NFHS 4 | NFHS 5 | NFHS 4 | NFHS 5 |
|---|---|---|---|---|---|---|
| | Diabetes | Diabetes | Hypertension | Hypertension | Both | Both |
| | aOR (95% CI) | aOR (95% CI) | aOR (95% CI) | aOR (95% CI) | aOR (95% CI) | aOR (95% CI) |
| **Male/Female** | | | | | | |
| 15–19 | 1 | 1 | 1 | 1 | 1 | 1 |
| 20–24 | 1.32* (1.17–1.49) | 1.07 (0.98–1.16) | 1.6* (1.54–1.66) | 1.43* (1.37–1.49) | 2.17* (1.41–3.35) | 1.20 (0.93–1.55) |
| 25–29 | 1.96* (1.73–2.21) | 1.07* (1.20–1.43) | 2.29* (2.21–2.39) | 2.02* (1.94–2.11) | 5.36* (3.58–8.03) | 1.77* (1.38–2.26) |
| 30–34 | 3.20* (2.84–3.61) | 1.83* (1.67–2.00) | 3.36* (3.23–3.51) | 2.98* (1.94–2.11) | 11.89* (8.00–17.69) | 3.35* (2.63–4.26) |
| 35–39 | 5.14* (4.57–5.79) | 2.86* (2.62–3.13) | 4.75* (4.56–4.96) | 4.32* (4.14–4.50) | 24.55* (16.57–36.37) | 6.83* (5.40–8.64) |
| 40–44 | 8.66* (7.70–9.75) | 4.32* (3.96–4.72) | 6.43* (6.17–6.71) | 6.25* (6.00–6.52) | 47.42* (32.06–70.15) | 11.93* (9.45–15.05) |
| 45–49 | 13.07* (11.63–14.7) | 6.33* (5.81–6.90) | 8.50* (8.15–8.87) | 8.40* (8.06–8.76) | 50.54* (42.03–63.72) | 19.52* (15.49–24.60) |
| >50 | 20.94* (18.30–23.96 | 10.05* (9.02–11.2) | 11.53* (10.85–12.27) | 12.76* (11.98–13.59) | 65.32* (52.26–72.63) | 35.37* (27.47–45.53) |
| **BMI** | | | | | | |
| <18 | 1.00 | 1.00 | 1.00 | 1.00 | 1.00 | 1.00 |
| 18.5–22.9 | 1.30* (1.22–1.38) | 1.21* (1.15–1.28) | 1.28* (1.26- | 1.16* (1.22–1.19) | 1.93* (1.66–2.25) | 1.53* (1.35–1.74) |
| 23.0–24.9 | 1.99* (1.86–2.13) | 1.66* (1.57–1.76) | 1.92* (1.87- | 1.64* (1.59–1.68) | 3.96* (3.39–4.62) | 2.72* (2.39–3.10) |
| 25.0–27.4 | 2.95* (2.76–3.16) | 2.19* (2.06–2.32) | 2.64* (2.57- | 2.16* (2.10–2.22) | 6.66* (5.72–7.74) | 4.17* (3.67–4.74) |
| 27.5–29.9 | 3.94* (3.66–4.23) | 2.75* (2.58–2.93) | 3.36* (3.25- | 2.75* (2.66–2.84) | 10.21* (8.75–11.90) | 5.95* (5.21_6.79) |
| >30 | 5.77* (5.39–6.19) | 4.06* (3.82–4.32) | 4.40* (4.26- | 3.55* (3.44–3.67) | 15.88* (13.66–18.47) | 9.81* (8.63–11.15) |
| **Marital Status** | | | | | | |
| Never | 1.00 | 1.00 | 1.00 | 1.00 | 1.00 | 1.00 |
| Married | 1.01 (0.94–1.10) | 1.09 (0.94–1.26) | 0.85 (0.82- | 0.91 (0.88–0.94) | 0.86 (0.73–1.02) | 1.11* (1.05–1.19) |
| Widowed | 1.06 (0.95–1.18) | 1.23* (1.03–1.46) | 0.91 (0.86- | 1.03 (0.99–1.08) | 0.99 (0.82–1.21) | 1.16* (1.06–1.27) |
| Divorced | 0.95 (0.75–1.21) | 1.24 (0.87–1.75) | 0.98 (0.88- | 0.94 (0.84–1.04) | 0.81 (0.53–1.22) | 1.20 (0.98–1.47) |
| No longer living together | 1.27* (1.05–1.52) | 1.07 (0.80–1.43) | 0.78 (0.71- | 0.95 (0.88–1.03) | 0.77 (0.53–1.10) | 1.19* (1.02–1.39) |
| **Education** | | | | | | |
| No Education | 1.00 | 1.00 | 1.00 | 1.00 | 1.00 | 1.00 |
| Incomplete Primary | 1.31* (1.22–1.40) | 1.30* (1.24–1.35) | 1.14* (1.11- | 1.11* (1.08–1.13) | 1.33* (1.19–1.48) | 1.47* (1.37–1.58) |
| complete Primary | 1.29* (1.21–1.38) | - | 1.01 (0.98- | - | 1.32* (1.19–1.46) | - |
| Incomplete Secondary | 1.50* (1.44–1.57) | 1.50* (1.45–1.55) | 1.05* (1.03- | 1.11* (1.09–1.13) | 1.42* (1.33–1.52) | 1.62* (1.53–1.71) |
| complete Secondary | 1.46* (1.37–1.56) | 1.37* (1.23–1.51) | 0.98 (0.95- | 1.09* (1.04–1.15) | 1.31* (1.17–1.47) | 1.42* (1.16–1.72) |

*(Continued)*

**Table 5.** (Continued)

|  | NFHS 4 | NFHS 5 | NFHS 4 | NFHS 5 | NFHS 4 | NFHS 5 |
|---|---|---|---|---|---|---|
|  | Diabetes | Diabetes | Hypertension | Hypertension | Both | Both |
|  | aOR (95% CI) | aOR (95% CI) | aOR (95% CI) | aOR (95% CI) | aOR (95% CI) | aOR (95% CI) |
| Higher | 1.47* (1.39–1.56) | 1.37* (1.30–1.44) | 0.90 (0.87- | 1.00 (0.97–1.02) | 1.34* (1.22–1.48) | 1.40* (1.29–1.53) |

subsequently explained using other large cohort studies how HTN uncontrolled blood pressure promotes insulin resistance through known and unknown pathways to cause DM. This mechanism is unclear, but alarm bells must be rung to emphasise this relation as we uncover it.

If we don't take timely measures to monitor the prevalence of DM and HTN across different states in India, we risk overlooking cases and ending up with a larger number of individuals requiring treatment for both conditions, which could lead to a heavier burden on our economy. Given India's vast size, it is crucial that we focus on this issue to prevent such outcomes.

## 6. Role of Ayushman Bharat and Health Wellness Centres as way forward

As evidenced for best approach, we need a comprehensive package which focuses on early diagnosis and treatment of both HTN and DM. With the introduction of technology in the form of digital health records, the government's Ayushman Bharat commitment is poised to provide a comprehensive solution for Noncommunicable Diseases (NCDs). This program anticipates delivering primary care services through Health Wellness Care Centres (HWCs) [28, 29]. Our results can be utilised for prioritizing the resources as per prevalence in various existing channels in Ayushman Bharat program which is now being integrated in Ayushman Bharat Digital Mission (ABDM) [30]. ABDM will provide more real time data and can help immensely trained and skilled health care force to make early inroads in preventing the rise of HTN and DM. An integrated approach amalgamating the three existing frameworks of AB, ABDM and HCWs with right approach and guidance can definitely solve this increasing menace of concurrent rise of DM and HTN.

## Strengths of study

The study's strengths included a large, nationally representative survey dataset and a systematic analysis procedure. Our findings apply to more regions. Second, a comprehensive sample analysis design was used to consider sampling units and weighting. In addition, the study identified the high-risk group and the Indian states with the lowest blood pressure and glucose screening rates. Third We compared three-year-old country data bases. Trends from such a vast data set can forecast future events in India. We followed the definition of HTN as per the ESC guidelines for hypertension and its various operational definition [31].

## Conclusion

As can be seen in the comparative trend the overall prevalence of chronic diseases in Indian states and population has risen over the years. This trend follows an upward slope. The worrisome part is the rise in individuals having the concurrent presence of DM and HTN. This mandates more research for such individuals and more extensive screening for looking out for such individuals. Cross-referral form cardiac screening /treatment clinics for DM screening

and vice versa should be the norm rather than necessity. Resource allocation for early diagnosis of DM, HTN and their concurrent presence can be done on the basis of prevalence in various states. Public needs to be made aware about the various threats these two major non communicable diseases pose to them. A careful integration of existing infrastructure in form of ABDM and HCWs can help solving the rising menace.

## Limitations of the study

Since only data from females aged 15–50 was available, this study cannot reflect the entire population. NFHS 4 and 5 prioritised maternal and child health, hence male samples were smaller than female samples. Data from >2 lakh males is sufficient for country-specific estimates. We used blood glucose levels from the data set and cannot tell if the patient had type 2 or type 1 DM. NFHS-4 survey data lacks waist hip ratio data.

## Supporting information

**S1 File. This file contains all supporting figures and tables for the manuscript.** They are numbered with captions as follows: Fig 1. Prevalence of DM (% wise) as per NFHS 4 and 5 in States across India. Fig 2. Prevalence of HTN across states (% wise) across India. Fig 3. Prevalence of both DM and HTN in same individuals (% wise) across the states in India.Fig 4. Self-Reported Disorders in female as per NFHS-4 and NFHS-5. Fig 5. Self-Reported Disorders in male as per NFHS-4 and NFHS-5. Table 1. NFHS-5 All means compared with ANOVA. Table 2. NFHS 4 All means compared with ANOVA. Table 3. Multivariate Regression analysis on Self-Reported Disorders.
(DOCX)

## Acknowledgments

The authors would like to acknowledge and thank Resource Center for Tobacco control in India (RCTC) established under the Postgraduate Institute of Medical Education and Research (PGIMER), Chandigarh for providing technical support towards writing the manuscript. We are also grateful to Global Health Advocacy Incubator (GHAI) for supporting the study and Demographic and Health Surveys (DHS) Program for providing the data set (survey ref no. 155509 downloaded on June 3, 2022) which helped in the development of the manuscript.

## Author Contributions

**Conceptualization:** Rishabh Kumar Rana, Sitanshu Sekhar Kar, Sonu Goel.

**Data curation:** Rishabh Kumar Rana, Ravi Ranjan Jha, Ratnesh Sinha, Dewesh Kumar, Richa Jaiswal, Urvish Patel, Jang Bahadur Prasad.

**Formal analysis:** Rishabh Kumar Rana, Ravi Ranjan Jha, Ratnesh Sinha, Jang Bahadur Prasad.

**Funding acquisition:** Ratnesh Sinha, Sitanshu Sekhar Kar.

**Investigation:** Rishabh Kumar Rana, Sitanshu Sekhar Kar.

**Methodology:** Rishabh Kumar Rana, Dewesh Kumar, Jang Bahadur Prasad.

**Project administration:** Sitanshu Sekhar Kar, Sonu Goel.

**Resources:** Ratnesh Sinha, Richa Jaiswal.

**Software:** Richa Jaiswal, Urvish Patel.

**Supervision:** Rishabh Kumar Rana, Ravi Ranjan Jha, Ratnesh Sinha, Dewesh Kumar.

**Validation:** Ravi Ranjan Jha, Ratnesh Sinha, Sonu Goel.

**Visualization:** Dewesh Kumar, Richa Jaiswal, Urvish Patel, Sonu Goel.

**Writing – original draft:** Rishabh Kumar Rana, Ratnesh Sinha.

**Writing – review & editing:** Rishabh Kumar Rana, Dewesh Kumar, Richa Jaiswal, Urvish Patel, Jang Bahadur Prasad, Sitanshu Sekhar Kar.

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
