## [Decision Letter · Decision Letter 0]

8 May 2024

PONE-D-23-28478Correlates of Diabetes Mellitus and Hypertension in India: Change as evidenced from NFHS- 4 and 5 during 2015-2021.PLOS ONE

Dear Dr. Goel,

Thank you for submitting your manuscript to PLOS ONE. After careful consideration, we feel that it has merit but does not fully meet PLOS ONE’s publication criteria as it currently stands. Therefore, we invite you to submit a revised version of the manuscript that addresses the points raised during the review process.

We look forward to receiving your revised manuscript.

Kind regards,

Mohammad Hifz Ur Rahman

Academic Editor

PLOS ONE

4. Please include a caption for figure 1 and 2.

Authors are requested to address the reviewers comment and submit the revised version of the manuscript.

Reviewers' comments:

Reviewer's Responses to Questions

**Comments to the Author**

1. Is the manuscript technically sound, and do the data support the conclusions?

Reviewer #1: Yes

Reviewer #2: Yes

2. Has the statistical analysis been performed appropriately and rigorously? 

Reviewer #1: Yes

Reviewer #2: Yes

3. Have the authors made all data underlying the findings in their manuscript fully available?

Reviewer #1: Yes

Reviewer #2: Yes

4. Is the manuscript presented in an intelligible fashion and written in standard English?

Reviewer #1: Yes

Reviewer #2: Yes

5. Review Comments to the Author

Reviewer #1: 1. Mention the brief methodology in the abstract as well. How was the data sourced can be mentioned.

2. Start the sentences with words only and avoid numbers wherever possible.

3. The outline about the NFHS in the background section should be improved with respect to the writing style used.

4. The methodology should be restricted to the methods used in the current study only and not for NFHS or DLHS. The methodology used for the surveys can be given in the supplementary articles/appendix.

5. Results and discussion are well written but would benefit for citation of some more recent references such as

a. Basu S, Maheshwari V, Gokalani R, Lahariya C. Prevalence and predictors of gestational diabetes mellitus and overt diabetes in pregnancy: A secondary analysis of nationwide data from India. Prev Med Res Rev 2024;1:52-8. DOI: 10.4103/PMRR. PMRR_11_23

b. Shivani, Sajjan1; Chawla, Parvinder S.1; Kalra, Khyati1. Is the Rule of Halves Still Relevant today? A Study from a Tertiary Care Hospital in India. Preventive Medicine Research & Reviews 1(3):p 134-136, May–Jun 2024. | DOI: 10.4103/PMRR.PMRR_66_23

6. The authors also need to contextualise the findings with Ayushman Bharat programs and packages under HWC on how these findings can be tackled through these programs and cite the relevant papers such as

a. Lahariya C. Health & Wellness Centers to Strengthen Primary Health Care in India: Concept, Progress and Ways Forward. Indian J Pediatr. 2020; 87: 916-29. doi: 10.1007/s12098-020-03359-z. Epub 2020 Jul 8.

b. Lahariya C. 'Ayushman Bharat' Program and Universal Health Coverage in India. Indian Pediatr. 2018; 55: 495-506.

Reviewer #2: Dear Authors, the manuscript is meticulously written and has included all the required points. The paper has strong methodology and statistical approach. The data of NFHS 4 and NFHS 5 have been correlated in context to DM and HTN which seems to be very relevant.

6. PLOS authors have the option to publish the peer review history of their article (what does this mean?). If published, this will include your full peer review and any attached files.

Reviewer #1: No

Reviewer #2: **Yes: **Medha Mathur

---

## [Author Response · Author response to Decision Letter 0]

16 May 2024

Response to Reviewer - 

Thanks for reviewing our manuscript and suggestions to improve it further : 

Comments from-

Reviewer 1. A line-by-line response has been detailed below.

Comments received Change Line no Remarks/Response

1. Mention the brief methodology in the abstract as well. How was the data sourced can be mentioned. Changed 52 

2. Start the sentences with words only and avoid numbers wherever possible. Changed 96-97 Sentence has been reframed as per the suggestion

3. The outline about the NFHS in the background section should be improved with respect to the writing style used. Changed 96-104 Section has been rephrased 

4. The methodology should be restricted to the methods used in the current study only and not for NFHS or DLHS. The methodology used for the surveys can be given in the supplementary articles/appendix. Changed 132-136 Methods section describing the details have been rewritten keeping it brief .

5. Results and discussion are well written but would benefit for citation of some more recent references such as 

a. Basu S, Maheshwari V, Gokalani R, Lahariya C. Prevalence and predictors of gestational diabetes mellitus and overt diabetes in pregnancy: A secondary analysis of nationwide data from India. Prev Med Res Rev 2024;1:52-8. DOI: 10.4103/PMRR. PMRR_11_23

b. Shivani, Sajjan1; Chawla, Parvinder S.1; Kalra, Khyati1. Is the Rule of Halves Still Relevant today? A Study from a Tertiary Care Hospital in India. Preventive Medicine Research & Reviews 1(3):p 134-136, May–Jun 2024. | DOI: 10.4103/PMRR.PMRR_66_23 changed but only one citation has been used of Gestational diabetes . 352 Other ref is only from a tertiary care hospital and it would not be justified to use that for a large data set . We have used a similar reference in national context 

6. The authors also need to contextualise the findings with Ayushman Bharat programs and packages under HWC on how these findings can be tackled through these programs and cite the relevant papers such as

a. Lahariya C. Health & Wellness Centers to Strengthen Primary Health Care in India: Concept, Progress and Ways Forward. Indian J Pediatr. 2020; 87: 916-29. doi: 10.1007/s12098-020-03359-z. Epub 2020 Jul 8.

b. Lahariya C. 'Ayushman Bharat' Program and Universal Health Coverage in India. Indian Pediatr. 2018; 55: 495-506. Used and rewrote one section of the manuscript to utilise the references 373-385 

Reviewer #2: Dear Authors, the manuscript is meticulously written and has included all the required points. The paper has strong methodology and statistical approach. The data of NFHS 4 and NFHS 5 have been correlated in context to DM and HTN which seems to be very relevant.

Response – Thanks for the kind words and doing the review .

---

## [Decision Letter · Decision Letter 1]

28 May 2024

Correlates of Diabetes Mellitus and Hypertension in India: Change as evidenced from NFHS- 4 and 5 during 2015-2021.

PONE-D-23-28478R1

Dear Dr. Goel

We’re pleased to inform you that your manuscript has been judged scientifically suitable for publication and will be formally accepted for publication once it meets all outstanding technical requirements.

Kind regards,

Mohammad Hifz Ur Rahman

Academic Editor

PLOS ONE

Additional Editor Comments (optional):

Reviewers' comments:

Reviewer's Responses to Questions

**Comments to the Author**

1. If the authors have adequately addressed your comments raised in a previous round of review and you feel that this manuscript is now acceptable for publication, you may indicate that here to bypass the “Comments to the Author” section, enter your conflict of interest statement in the “Confidential to Editor” section, and submit your "Accept" recommendation.

Reviewer #1: All comments have been addressed

2. Is the manuscript technically sound, and do the data support the conclusions?

Reviewer #1: Yes

3. Has the statistical analysis been performed appropriately and rigorously? 

Reviewer #1: Yes

4. Have the authors made all data underlying the findings in their manuscript fully available?

Reviewer #1: Yes

5. Is the manuscript presented in an intelligible fashion and written in standard English?

Reviewer #1: Yes

6. Review Comments to the Author

Reviewer #1: All required comments have been addressed.

This can be considered for publication by the journal

7. PLOS authors have the option to publish the peer review history of their article (what does this mean?). If published, this will include your full peer review and any attached files.

Reviewer #1: **Yes: **Chandrakant Lahariya

---

## [Editor Report · Acceptance letter]

13 Jun 2024

PONE-D-23-28478R1 

PLOS ONE

Dear Dr. Goel, 

I'm pleased to inform you that your manuscript has been deemed suitable for publication in PLOS ONE. Congratulations! Your manuscript is now being handed over to our production team.

Kind regards, 

on behalf of

Dr. Mohammad Hifz Ur Rahman 

Academic Editor

PLOS ONE